# Needs and Research Priorities for Young People with Spinal Cord Lesion or Spina Bifida and Their Caregivers: A National Survey in Switzerland within the PEPSCI Collaboration

**DOI:** 10.3390/children9030318

**Published:** 2022-02-27

**Authors:** Irina Benninger, Patricia Lampart, Gabi Mueller, Marika Augutis, Inge Eriks-Hoogland, Sebastian Grunt, Erin Hayes Kelly, Beth Padden, Cordula Scherer, Sandra Shavit, Julian Taylor, Erich Rutz, Anke Scheel-Sailer

**Affiliations:** 1Swiss Paraplegic Center, Guido A. Zäch Strasse 1, 6207 Nottwil, Switzerland; irina.benninger@gmail.com (I.B.); patricia.lampart@paraplegie.ch (P.L.); gabi.mueller@paraplegie.ch (G.M.); inge.eriks@paraplegie.ch (I.E.-H.); 2Department of Neurobiology, Care Sciences and Society, Division of Neurogeriatrics, Karolinska Institutet, Solnavägen 1, 171 77 Stockholm, Sweden; marika.augutis@rvn.se; 3Department of Pediatrics, Division of Child Neurology, University Children’s Hospital Bern, University of Bern, Freiburgstrasse 15, 3010 Bern, Switzerland; Sebastian.Grunt@insel.ch; 4American Academy of Pediatrics, 345 Park Boulevard, Itasca, IL 60143, USA; ekelly@aap.org; 5Pediatric Rehabilitation, Center for Spina Bifida, University Children's Hospital Zurich, Steinwiesstrasse 75, 8032 Zürich, Switzerland; Beth.Padden@kispi.uzh.ch; 6Department of Pediatric Surgery, Children’s Hospital Bern, Freiburgstrasse 15, 3010 Bern, Switzerland; Cordula.Scherer@insel.ch; 7Department of Pediatric Surgery, Children's Hospital Lucerne, Spitalstrasse, 6000 Lucerne, Switzerland; sandra.shavit@luks.ch; 8National Spinal Injuries Centre and Stoke Mandeville Spinal Research, Buckinghamshire Healthcare NHS Trust, Aylesbury, Amersham HP7 0JD, UK; juliantaylorgreen2@gmail.com; 9Department of Orthopaedics, The Royal Children’s Hospital, Melbourne 3052, Australia; erich_rutz@hotmail.com; 10Medical Faculty, University of Basel, 4001 Basel, Switzerland; 11Department of Paediatrics, The University of Melbourne, Melbourne 3010, Australia; 12Murdoch Children’s Research Institute, Melbourne 3052, Australia

**Keywords:** spina bifida, spinal cord injury/disorder, quality of life, user perspective, research priorities

## Abstract

The aim of this study was to describe the needs and research priorities of Swiss children/adolescents and young adults (from here, “young people”) with spinal cord injury/disorder (SCI/D) or spina bifida (SB) and their parents in the health and life domains as part of the international Pan-European Pediatric Spinal Cord Injury (PEPSCI) collaboration. Surveys included queries about the satisfaction, importance, research priorities, quality of life (QoL), and characteristics of the young people. Fifty-three surveys with corresponding parent-proxy reports were collected between April and November 2019. The self-report QoL sum scores from young people with SCI/D and SB were 77% and 73%, respectively. Parent-proxy report QoL sum scores were lower, with 70% scores for parents of young people with SCI/D and 64% scores for parents of young people with SB. “Having fun”, “relation to family members”, and “physical functioning” were found to be highly important for all young people. “Physical functioning”, “prevention of pressure injuries”, “general health”, and “bowel management” received the highest scores for research priority in at least one of the subgroups. As parents tend to underestimate the QoL of their children and young people prioritized research topics differently, both young peoples’ and caregivers’ perspectives should be included in the selection of research topics.

## 1. Introduction

Recently, rehabilitation philosophy has become more patient-orientated. Studies have shown that patient and public involvement can have positive impacts on research by enhancing its quality and ensuring its appropriateness and relevance [1]. Consistency between research and consumer priorities and expectations should be a goal for practitioners and could ultimately improve rehabilitation outcomes [2,3].

Pediatric rehabilitation is complex. It faces not only the age-specific needs of patients but also the needs of parents and caregivers. In pediatric rehabilitation, children, adolescents, and young adults (from here, “young people”) with spinal cord injury/disorder (SCI/D) and spina bifida (SB) represent a small group living with relevant and lifelong impairments, sharing, to some extent, comparable needs [4,5]. Pediatric SCI/D is a very rare health condition [6,7,8] causing challenges to physical and psychosocial health, with an estimated number of new traumatic cases of 7–13 per year in the 0–14 age group in Switzerland [9]. The prevalence of SB at birth is 19.4 cases per 100,000 live births in Switzerland (Federal Office for statistics, 2014). Although small in number, these young people experience significant challenges that need to be addressed.

Young people with SCI/D and SB have to deal with and adapt to several SCI/D- or SB-associated impairments, growing-related challenges, and secondary health conditions. The SCI/D or SB-related physiological changes include sensory and motor deficits; autonomic dysregulation (including respiratory, cardiac, and circulatory dysfunction); and bladder, bowel, and sexual dysfunction. [10,11,12]. SCI/D- or SB-related complications in young people mainly include constipation, pressure injuries, spasticity, neuropathic pain, and urinary tract infections due to their complicated bladder management [13]. In addition to adults, young people with SCI/D or SB have to cope with nutrition intake and weight adaptation during the growing phase, with the risk of becoming underweight or obese (due to lower resting metabolic rates and decreased muscle mass) [14,15], and they suffer from an increased risk of developing hip instability (dislocation and subluxation) and scoliosis (due to their lack of trunk muscles) [16]. Young people with SB often face additional needs—for example, cognitive impairments due to Arnold Chiari malformation and hydrocephalus [4,5] or urological dysfunction and hip malformation since birth [17,18,19,20].

Further, living with a disability as a young person can also lead to psychosocial issues, such as depression and anxiety, which may influence school integration, education, and participation [8,10,11,21,22]. The above-mentioned issues can have an effect on the overall quality of life (QoL), independence, psychosocial health, and outcomes relating to participation and coping [8,11,23]. Given the variety of complications faced by young people with SCI/D or SB, how to comprehensively prioritize and address these needs remains unclear.

In 2012, a group of European rehabilitation specialists with an interest in pediatric SCI/D rehabilitation formed the Pan-European Pediatric Spinal Cord Injury (PEPSCI) collaboration. The overall aim of this collaboration was to collect data on health conditions, social and emotional situations, and the school integration patterns of young people with SCI/D [24] in Europe. As part of this international project, the purpose of this national study was to assess the importance, satisfaction, and research relevance in the health and life domains as well as QoL among young people with SCI/D and SB and their parents in Switzerland.

As young people with SCI/D and SB sometimes participate in rehabilitation programs together [25,26], understanding the various needs and priorities of both patients with SCI/D and patients with SB can offer guidance on how to tailor rehabilitation programs to meet their needs.

## 2. Materials and Methods

### 2.1. Study Design

This is a descriptive study based upon a cross-sectional survey in the German-speaking part of Switzerland as part of an international collaboration (PEPSCI).

### 2.2. Patients and Setting

Eligible young people with SCI/D or SB aged from 2 to 25 living in the German-speaking part of Switzerland and their parents were included. The participants were tracked down in five large hospitals treating children, youths, and young adults with SCI/D and/or SB: children’s hospitals in Basel, Berne, Lucerne, and Zürich and the Swiss Paraplegic Center. In the case of acquired SCI/D, the date of the injury had to be before the patient’s 18th birthday. Participants with severe neurological deficits (e.g., because of acquired brain injury) or with no need of any assistive technologies were excluded.

### 2.3. Screening and Recruitment

Young people with SCI/D or SB were identified from the departmental or institutional databases of each participating institute. The standard procedure was for local collaborators to identify all the young people with SCI/D and SB. The screening of the inclusion and exclusion criteria was based on medical information in the institutional database. The eligible participants and their parents received the questionnaires by postal mail, including an information letter and informed consent form. The informed consent form was either signed by parents and participants or just parents depending on the age of the participant (Appendix A). According to the Swiss ethics guidelines, children until the age of ten were orally informed about the study by their parents. From an age of eleven onward, children received an age-adapted participant information form but did not sign it themselves. From an age of fourteen onward, children received an age-adapted participant information form and signed the form themselves. In some institutions, the potential participants were additionally informed about the study by telephone by their health professionals. The survey was sent in 2019 in the German-speaking part of Switzerland.

One to three months after sending the questionnaires, the young people/parents who did not return the questionnaire were contacted by telephone. During this telephone call, the first author explained again the purpose of the study and reminded the individual that he or she was under no obligation to participate and that it would not influence their further treatment if he or she did not participate.

### 2.4. Development of Survey

The survey contains four parts: Part I, the basic information form; Part II, the PedsQL™ (pediatric quality of life questionnaire) [27]; Part III, the health and life domain questionnaire; and Part IV, a neurological form. The second and third part were organized as 3- to 5-point Likert scales, with an additional free text section in the third part (Appendix A). The different parts were adapted for different age groups (Appendix A).

#### 2.4.1. Part I—Basic Information Form

This questionnaire contained 12 items and aimed to obtain demographic information about the young people with pediatric SCI/D and SB (i.e., their age; gender; education level; and time of, cause of, and years since their injury). The basic information form was completed by the parents of young people aged 2 to 14, whereas young people aged 15 to 25 filled out the basic information form by themselves. An additional question regarding the education status of the parents was asked for all ages.

#### 2.4.2. Part II—PedsQL™

To describe the QoL, we used the German-translated validated PedsQL™ [27].

The PedsQL™ is a modular instrument used for measuring health-related QoL in children and adolescents from 2 to 18 years of age. The PedsQL™ Generic Core Scales are multidimensional child self-reports and parent-proxy report scales developed by J. W. Varni and associates over the last 15 years [27].

The Generic Core Scales included four functional domains: (1) physical (8 items), (2) emotional (5 items), (3) social (5 items), and (4) school functioning (5 items). Separate questionnaires were provided for young people between the ages of 2 and 4 (parent proxy report only), 5 and 7 (self and parent-proxy reports), 8 and 12 (self and parent-proxy reports), 13 and 17 (self and parent-proxy reports), and 18 to 25 (self and parent-proxy reports) with SCI/D or SB. The questionnaire for ages 5–7 years consisted of graphic 3-point Likert scales, whereas the questionnaires for the ages 8 to 25 years and all parent-proxy reports consisted of 5-point Likert scales. To be more inclusive to the SCI/D and SB population, slight modifications to the wording of the questions for physical functioning (mainly questions concerning walking ability) were made with Varni’s permission.

#### 2.4.3. Part III—Health and Life Domain Questionnaire (HLDQ)

PEPSCI collaborators developed this questionnaire based on the findings presented in Simpson and colleagues’ systematic review of the health and life priorities of adults with SCI/D and SB [3]. In this review, two central domains were identified: (1) the “life” domain and (2) the “health” domain. We decided to ask about (A) importance, (B) satisfaction, and (C) research priority in relation to Simpson's detected life and health domains. We started out by looking at the adult survey carried out in the UK [2]. After the questionnaire was developed in the expert committee, the questions were validated for understanding among children, adolescents, and parents in the UK, the US, and Sweden. Some adaptations were made concerning the wording and the description of sexuality. Separate questionnaires were provided for young people from 8 to 12, 13 to 17, and 18 to 25 years of age (all with corresponding parent-proxy reports). For young people younger than 8 years, questionnaires were only filled out by their parents as parent-proxy reports. We had to make some adaptions to age groups to meet the national ethical requirements in Switzerland.

A free text option was available for all participants, allowing to write about additional aspects that they would like SCI/D and SB researchers to focus on in the future.

#### 2.4.4. Part IV—Neurology Form

A baseline characteristic form was developed by the PEPSCI Collaboration, providing details about the individuals’ SCI/D and SB (gender, date of onset, level, completeness, and type of injury).

The first author completed this form.

### 2.5. Translation Process

We translated the HLDQ according to the cross-cultural translation guideline of Epstein and Sousa in two steps [28,29]:

Stage 1: Bilingual translators whose mother tongue was the target language produced the two independent translations. These translators did not need to be professional translators. Preferably, they were experts in the medical field. Translator 1 (T1) was aware of the concepts in rehabilitation medicine, SCI/D, SB, etc. Translator 2 (T2) was familiar with colloquial phrases, health care slang and jargon, idiomatic expressions, and emotional terms in common use.

Stage 2: The expert committee reviewed the quality and content of the translations produced by T1 and T2 and ensured that no meaning was lost. The Expert Committee consisted of all the translators (T1 and T2) and health care professionals involved up to this point. The composition of this expert committee was crucial for the achievement of cross-cultural equivalence.

### 2.6. Data Collection

Data collection was based on the returned paper-based questionnaires. For all who returned the questionnaire, the first author collected the medical information from the original medical records from each participating center. Furthermore, data were reviewed and verified prior to data entry completion. Data were entered into secuTrial^®^ (secure data server with no participant identifiers).

### 2.7. Statistical Analysis

Means and standard deviations (SD) or medians with 25th and 75th percentiles were calculated. The PedsQL™ questionnaire was recoded prior to analysis according to the following website: http://www.pedsql.org/score.html (accessed on 4 December 2018). Then, the sum scores were calculated for physical, emotional, social, and school functioning by adding all the corresponding scores and dividing them by the number of answered items. After this step, the mean sum scores for physical health (=physical functioning) and for psychosocial health (=mean of emotional, social, and school functioning), and the total score (mean of the four above-mentioned sum scores) were calculated. Scores for the PedsQL™ were given as medians with 25th and 75th percentiles for patients and parents, both of them subdivided into SCI/D and SB.

For the Health and Life Domain Questionnaire, the means for the health and life domains were calculated for each question of the corresponding subgroup. Missing answers were replaced by zero, as not filling in a question was interpreted as low importance and the number of missing answers for each question was added. The highest five values of importance, satisfaction, and research were shown and highlighted in gray in the tables. The means of lowest satisfaction were also displayed.

Microsoft Excel and PASW Statistics 18 were used for all analyses (SPSS Inc., Chicago, IL, USA).

## 3. Results

### 3.1. Study Population

In total, 185 young people with SCI/D (32) and SB (153) and their corresponding parents were eligible to participate. Of these, 53 surveys from young people with SCI/D and SB with corresponding parent-proxy reports were returned: 15 were from young people with SCI/D and 38 were from young people with SB. This led to a response rate of 28.6% in total, 46.9% for SCI/D and 24.8% for SB. In the SCI/D group, 12 surveys (out of 15, 80%) were filled out by boys, while in the SB group 24 (out of 38, 63%) were filled out by girls. The mean age at injury for the SCI/D population was 9.9 years, and most of the participants had a neurological level between Th1 and Th12, with it being considered that the neurological examination of children remained challenging but possible (Appendix A).

The education level of the parents differed in terms of percentages at university level (SCI/D 8% versus SB 28%). The percentages for primary school and vocational training were similar (Appendix A).

The SCI/D population composed of half traumatic origin and half non-traumatic origin patients (bleeding, transverse myelitis, congenital spinal cord lesions, and tumors) (Appendix A).

### 3.2. Quality of Life (PedsQL™)

The total sum score of the PedsQL™ ranged between 64.5% and 77.4%. The young people with SCI/D and SB had total QoL scores of 77.4% and 73.1%, respectively. Parent-proxy report scores were lower with 70, 2% for young people with SCI/D and 64, 5% for young people with SB. The Likert score for school functioning for young people with SB was 75% and 60% in the corresponding parent-proxy reports versus 81.3% for young people with SCI/D and 80% in corresponding parent-proxy reports. The psychosocial health sum score was highest for young people with SCI/D. The results of PedsQL™ varied between age groups in the SB group mainly due to the scores in school functioning (Appendix A).

### 3.3. Satisfaction, Importance and Research Priorities in Health and Life Domains

“Relationships with family members” and “physical functioning” were the two domains mentioned as being within the top five either for satisfaction or importance in all subgroups. However, these items did not receive high research priority in all subgroups. For the young people with SCI/D and SB, the “what you do to have fun” domain received the highest score for importance but was not ranked within the top five domains for research priorities and satisfaction. It was not ranked within the top five in one of the three categories in the parent-proxy reports. For young people with SCI/D and SB, “time playing with or hanging out with others” was ranked within the top five for importance. “Ability to concentrate and learn new things” was ranked within the top five for importance in parent-proxy reports (SCI/D and SB) and for young people with SCI/D but not for young people with SB. It was not ranked within the top five for research priorities in any of the subgroups. “Bladder management” was ranked within the top five for research priorities in parent-proxy reports (SB and SCI/D) and for young people with SB, but not for young people with SCI/D (Appendix A).

Scores in satisfaction for the “ability of moving feet and legs” were low for the young people with SCI/D. Mobility (“how easy it is to get where you need to go”) was ranked within the top five for research priorities in participants with SCI/D and SB. Accessibility (“accessibility of your child’s home”) was ranked within the top five for importance in parent-proxy reports in the SCI/D group and within the top five for satisfaction in parent-proxy reports in the SB group (Appendix A).

For the young people with SB and their parents, the presence of spasms and muscle jumping and how those could be controlled were not ranked within the top five for importance, satisfaction, nor research priorities, whereas it was ranked within the top five for research priorities in parents of young people with SCI/D. The presence of pressure injuries and how they could be prevented received top scores for research priorities for the young people with SCI/D and SB (Appendix A).

The presence and treatment of pain was ranked within the top five for research priorities for young people with SCI/D and SB and parents of participants with SCI/D, but not for parents of participants with SB (Appendix A).

In the additional free text section, young people and their parents mentioned many survey topics again and added some items. Parents added physical, emotional, and social aspects, whereas young people mainly added physical aspects. Additional new aspects were neurological recovery, stem cells, nerve transfers, electrical stimulation, and the healing of the spinal cord. It was also mentioned that surgical procedures should be better explained to children (not with medical terms) and that research should be performed on how children can be supported to mentally and physically recover from traumatizing operations and procedures (Appendix A).

## 4. Discussion

Data from our questionnaire-based study provide a better understanding of the current needs and research priorities for young people with SCI/D and SB and their parents. This is the first study to reveal health and life priorities from the perspectives of children, adolescents, and young adults, as well as their parents and caregivers. In our study, only parents and no caregivers participated, and for that reason we only used the term ‘parents’.

Participants clearly selected different topics of research priorities. This underlines the idea that even young people and parents can clearly define their needs and research priorities and shows that young people and their parents can be integrated into the selection of research activities. Besides biomedical research topics such as neurological recovery, bowel and bladder management, pain, pressure injuries, and mobility, the following topics were also prioritized for research by the participants: relationships with friends and family members, social activities, integration, and participation.

### 4.1. Characteristics of the Study Population

We found, in total, 185 young people with SCI/D or SB in the German-speaking part of Switzerland. This result confirms that SCI/D and SB are rare health conditions [9] and the need for specialized health care services and international collaboration [24]. We contacted the highly specialized SCI/D and SB centers and screened the hospital data bases. We assume that all young people with SCI/D or SB were hospitalized at one point in one of the centers. We did not check the official data bank of national statistics and we did not include the French- and Italian-speaking parts of Switzerland due to language restrictions. The gender representation in our study was in accordance with the expected distribution, with a predominance of male participants in the SCI/D population of 80% [7,10] and a female predominance in the SB population of 63.2% [19,30]. The education status of the parents in the SCI/D population represented the average in Switzerland. The higher educational level of the parents of young people with SB could be explained by the higher age at birth of the parents of children with SB [26,31].

### 4.2. Perception of Quality of Life

We found differences in total QoL scores between the young people and parent-proxy reports. This confirmed that, in general, parents tend to underestimate the health and psychological condition of their children [32,33].

The lower results achieved in school scores for the SB group could be explained by the fact that the majority of children with SB, especially those with hydrocephalus, have cognitive impairments and therefore struggle more in school or after school time than those with SCI/D. It has been shown in several studies that most children with SB have a lower IQ compared to their able-bodied peers [26,34].

We found that the perception of the overall QoL does not only depend on physical health.

When comparing the QoL scores in our population with published data collected from children without disabilities, we found our participants to report lower scores for physical health and the same scores for social and psychological health [27]. The results of our sample were, however similar on all dimensions when compared to published data collected from children with rheumatoid arthritis [35] and cancer [36].

### 4.3. Importance, Satisfaction and Research Priorities

In general, social, psychological, and physical aspects influence the lived experience of young people with disabilities and their families [3]. Our population also mentioned relationships with family members as one of the most important aspects of their lives. As QoL was defined as including physical and psychological aspects [37], researchers have recently suggested that social aspects should also be included in research [38,39]. Comparing importance, satisfaction, and research priorities, the scores for research priorities were generally lower compared to those for other aspects. It might be possible that the participants—in particular, the younger children—were overstrained by questions about research priorities [1]. We realized that importance, satisfaction, and research priorities are different constructs and are perceived as such by participants, and concluded that something may be really important to someone's wellbeing but does not require research.

Nevertheless, many of the mentioned aspects—for example, the presence of muscle spasms in participants with SCI/D, pain, and bowel and bladder management issues—were already part of the most important research topics and research plans for adults and could be specifically addressed in younger people [1]. Recent research concerning access to health services has been conducted utilizing new methodological concepts and health system interventions that can be used to optimize the satisfaction of young people and families living with SCI/D or SB [19,40]. The results obtained for mobility and accessibility showed that these domains play an important role in the young people’s and parents’ everyday lives. The free answers also indicated that parents in particular are aware of several topics covering specific as well as bio-psycho-social aspects in general.

All the data collected for this study were self-reported or parent-proxy reports and vulnerable to problems and biases connected with self- and proxy-report measures. Related to the low incidence of people with SCI/D or SB, we only detected 185 participants, and, of those, the response rate was about 30% on average, with a 50% higher response for the participants with SCI. Although we sent reminders and followed up with phone calls in the no-response cases, the response rate reduces the generalizability of these results.

The low number of participants, especially those with SCI/D, did not allow for complex statistical analyses, so some subgroup analyses were conducted for young people with SB only.

Therefore, the descriptions of different topics might be overestimated due to the low number of responses.

In particular, our sample group did not contain any individuals with SCI/D or SB younger than 3 years of age. This clearly shows the relevance of international multicenter studies in overcoming the challenge of low participation numbers.

As we found a diverse range of perspectives, we think that we did not miss highly relevant perspectives. However, upcoming studies should include different languages and cultural regions. Age-specific versions of questionnaires were generated to ensure that the participants were provided with age-appropriate and cognitively relevant questions. However, this means that some of our data were not directly comparable across all age categories and cultural regions. As the cognitive development of children differs between individuals, larger cohorts are necessary to address the different age-related needs and perspectives.

## 5. Conclusions

Social, physical, and psychological topics were viewed in slightly different ways in all subgroups regarding importance, satisfaction, and research relevance. Parents tend to underestimate the health and psychological condition of their children, so the perspectives of young people themselves should also be integrated when research topics are selected in the future. The highly ranked social participation topics, as well as physical functioning, pressure injury prevention, and bowl management, should be included in the prioritized research topics.

This knowledge will be of great importance, helping rehabilitation clinics and health services to optimize their care and develop guidelines based on patients’ and caregivers’ perspectives.

## Data Availability

Electronic data are stored by the corresponding author at the Swiss Paraplegic Centre. Source data and identification data are stored in a locked archive room in the Swiss Paraplegic Centre with limited and controlled access for a minimum of 15 years.

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
