# Peer review of "Needs and Research Priorities for Young People with Spinal Cord Lesion or Spina Bifida and Their Caregivers: A National Survey in Switzerland within the PEPSCI Collaboration"

_children, 2022, doi:10.3390/children9030318_

Round 1
Reviewer 1 Report
This is well-designed and well-conducted survey of young people with SCI/D or SB or their parents in German-speaking portion of Switzerland to address concerns and research priorities. The authors used mostly previously validated questionnaires. Their findings are similar to larger studies in the US.
Introduction - good review of the salient issues.
Methods - Well-described. It is nice to see this type of survey using validated questionnaires. At what ages were patients allowed to give informed consent without parental approval?
Results - well-described but I am unable to see Tables referenced. Only the ones in the supplement are available.
Discussion - good review of the literature and how the findings of this study fit in.
Author Response
- We thank the reviewer for this comment. It has been a pleasure to look at this topic from a care user's/ children's point of view.
- According ti the siwssethics_Kinder Chckliste v1.1 26.04.2018, children untio the age of ten were orally informed about the study by their parents/caregiver. From the age of 11 onward children received a child-adapted participant information form and signed the form themselves. We added a clarifying sentence in line 128-132
- It was difficult to include the tables to the template and keep the formatting as they were very large. We therefore included all tables in the supplementary material and changed the number of the tables accordingly.
- We thank the reviewer for this comment.
Reviewer 2 Report
Dear Authors,
Congrats for your great work.
Best regars.
Author Response
We thank the reviewer for this nice and valuable feedback.
This manuscript is a resubmission of an earlier submission. The following is a list of the peer review reports and author responses from that submission.
Round 1
Reviewer 1 Report
The authors present the results of a survey of all children with spinal cord injury (SCI) and spina bifida (SB) in German-speaking Switzerland. The stated purpose is to describe needs and research priorities.
The authors use a quality of life measure that is standard and widely used. They note the mean scores for children performing self-rating are higher than parent assessment. However, there is no statistical comparison provided. Adding a pairwise comparison of the PedsQL scores is a necessary component to this analysis.
The authors use a Health and Life Domain Questionnaire. It appears that they have developed this questionnaire themselves. There is insufficient information provided about how it was developed, if it is validated, how the questions were worded. Because of these limitations to the HLDQ, it is difficult to know how to interpret the findings.
There are free-text discussions of research priorities as well. But the authors do not explain how these are integrated into the HLDQ questions. Likewise, there are many free text answers provided. This raises the question about whether the HLDQ was sufficient to accomplish its purpose.
The authors note that there are 185 patients total in the entire country. If this is true, and they are able to sample from the entire population, that is a strength of the study. However, it is not clear how they ascertained the total population, and how they assured that no one was missed.
Author Response
The authors present the results of a survey of all children with spinal cord injury (SCI) and spina bifida (SB) in German-speaking Switzerland. The stated purpose is to describe needs and research priorities.
The authors use a quality of life measure that is standard and widely used. They note the mean scores for children performing self-rating are higher than parent assessment. However, there is no statistical comparison provided. Adding a pairwise comparison of the PedsQL scores is a necessary component to this analysis.
We thank the reviewer for this comment. A pairwise comparison is a meaningful next step in understanding the differences between children and caregivers even better. Due to the small sample size a comparison was not possible from a statistical point of view.
We refer to this limitation in line 402-406.
The authors use a Health and Life Domain Questionnaire. It appears that they have developed this questionnaire themselves. There is insufficient information provided about how it was developed, if it is validated, how the questions were worded. Because of these limitations to the HLDQ, it is difficult to know how to interpret the findings.
We thank the reviewer for this remark.
We now explained the development and testing in more detail. It now reads: "We started out by looking at the adult survey made in UK (2). After the questionnaire was developed in the expert committee the questions were validated for understanding among children, adolescents and p/c in UK, US and Sweden. Some adaptation were made concerning wording and the description of sexuality." (Line 161-165)
There are free-text discussions of research priorities as well. But the authors do not explain how these are integrated into the HLDQ questions. Likewise, there are many free text answers provided. This raises the question about whether the HLDQ was sufficient to accomplish its purpose.
We thank the reviewer for this aspect. We discussed the advantage of using a questionnaire or performing a qualitative survey in advance. Using the questionnaire we added a short free part and summarized the results (Line 321-327).
We added the following sentence to the discussion part: The free answers also indicated that especially p/c are aware of several topics covering specific as well as bio-psycho-social aspects in general. (Line 391-392).
The authors note that there are 185 patients total in the entire country. If this is true, and they are able to sample from the entire population, that is a strength of the study. However, it is not clear how they ascertained the total population, and how they assured that no one was missed.
We thank the reviewer for this clarification.
We included the following explanation in the discussion for the recruitment process and the representativeness of the population.
It now reads: We contacted the highly specialized SCI and SB centers and screened the hospital data bank. We guess that all children with a SCI or a SB were hospitalized at one time point in one of the centers. Nevertheless we did not check the official data bank of the national statistics and we did not include the French and Italian speaking part of Switzerland. (Line 347-351)

Reviewer 2 Report
This paper addresses the importance of direct patient input in creating research initiatives.
Authors are cognizant of the limitations of conducting this survey in a single country where local culture, laws and infrastructure may not reflect the global attitudes and conditions for individuals with SB and SCI.
Authors reported PEDS QL scores in descriptive statistics. It may be helpful to present these scores with some context of PEDS QL scores reported in other studies in the same population and comparison with PEDS QL scores in other chronic conditions. Without that it it difficult to ascribe any meaning to the scores obtained.
Minor: line 51 would change finally to "ultimately"
Author Response
This paper addresses the importance of direct patient input in creating research initiatives.
Authors are cognizant of the limitations of conducting this survey in a single country where local culture, laws and infrastructure may not reflect the global attitudes and conditions for individuals with SB and SCI.
We thank the reviewer. We only included a national cohort with only German speaking participants according to our recruitment process. Therefore we recommend to add international aspects and compare our results with other countries.
We included the aspect of culture in the limitation section. It now reads: However, upcoming studies should include different languages and cultural regions. Age-specific versions questionnaires were generated, to ensure that participants were provided with age-appropriate and cognitively relevant questions. However, this means that some of our data were not directly comparable across all age categories and cultural regions. (Line 410-413)
Authors reported PEDS QL scores in descriptive statistics. It may be helpful to present these scores with some context of PEDS QL scores reported in other studies in the same population and comparison with PEDS QL scores in other chronic conditions. Without that it is difficult to ascribe any meaning to the scores obtained.
We thank the reviewer for this comment. We included a comparison with other patient group in our discussion part. It now reads: When comparing the QoL scores in our population with healthy children we found lower scores for physical health and the same scores for social and psychological health [27]. The results of our population were the same in all dimensions compared to a children population with rheumatoid arthritis [35] and cancer [36]. (Line 369-372)
Minor: line 51 would change finally to "ultimately"
We thank the reviewer for this recommendation and changed the word. It now reads: and could ultimately improve rehabilitation outcomes (Line 51)

Round 2
Reviewer 1 Report
The authors have made some minor changes that are responsive to reviewer critiques. However, much more substantial changes would be required for this to be appropriate for publication.
Author Response
Reviewers' comments:
Reviewer #1:
The authors have made some minor changes that are responsive to reviewer critiques. However, much more substantial changes would be required for this to be appropriate for publication.
We thank the reviewer for this feedback. We revised the whole manuscript and optimized the flow and the English. We clarified the children, youth and young adult perspective and added the parents and caregiver perspective. We highlighted the main results and discussed the conclusions for research activities. So we think that after all these revision the manuscript is relevant and add new aspects.
Attached you find a track changed manuscript and a clean version.
We confirm that all authors had substantial participation in the conception and design of the work, execution of the work, analysis of the data, contribution of methodological expertise. All authors also contributed to the writing of the manuscript and approved of the final version. The authors declare no conflicts of interest. We also confirm that the manuscript is not published elsewhere.
Sincerely,
Irina Benninger
Anke Scheel-Sailer
Erich Rutz
